# Posture Detection of Individual Pigs Based on Lightweight Convolution Neural Networks and Efficient Channel-Wise Attention

**DOI:** 10.3390/s21248369

**Published:** 2021-12-15

**Authors:** Yizhi Luo, Zhixiong Zeng, Huazhong Lu, Enli Lv

**Affiliations:** 1College of Engineering, South China Agricultural University, Guangzhou 510642, China; 20191009009@stu.scau.edu.cn (Y.L.); enlilv@scau.edu.cn (E.L.); 2Guangdong Academy of Agricultural Sciences, Guangzhou 510640, China; huazlu@scau.edu.cn

**Keywords:** pig postures, real-time detection, lightweight model, channel-wise attention

## Abstract

In this paper, a lightweight channel-wise attention model is proposed for the real-time detection of five representative pig postures: standing, lying on the belly, lying on the side, sitting, and mounting. An optimized compressed block with symmetrical structure is proposed based on model structure and parameter statistics, and the efficient channel attention modules are considered as a channel-wise mechanism to improve the model architecture.The results show that the algorithm’s average precision in detecting standing, lying on the belly, lying on the side, sitting, and mounting is 97.7%, 95.2%, 95.7%, 87.5%, and 84.1%, respectively, and the speed of inference is around 63 ms (CPU = i7, RAM = 8G) per postures image. Compared with state-of-the-art models (ResNet50, Darknet53, CSPDarknet53, MobileNetV3-Large, and MobileNetV3-Small), the proposed model has fewer model parameters and lower computation complexity. The statistical results of the postures (with continuous 24 h monitoring) show that some pigs will eat in the early morning, and the peak of the pig’s feeding appears after the input of new feed, which reflects the health of the pig herd for farmers.

## 1. Introduction

Pig postures reflect health status [1,2], welfare level [3], and environmental status [4]. Timely recognition of abnormal posture can help alleviate the spread of disease in pigs, reduce the use of veterinary antibiotics, and improve the economic benefits of commercial pig farms. The current pig industry requires farmers to obtain real-time information on the status of pigs for timely identification of individual pig abnormalities, but monitoring and evaluating an individual pig’s posture during its life on an intensive pig farm is hard for farmers [5,6]. With progress in image analysis technology, deep learning methods involve no-contact, are cheap, accurate, and have been widely used in tasks, such as animal detection and tracking [7,8], plant maturity, disease detection [9,10,11], and agricultural robotics [12,13,14,15]. Therefore, computer vision technology has wide prospects in regard to posture detection of individual pigs.

In regard to pig behavior detection—tail biting, crawling, and fighting behaviors, especially those associated with health and welfare and barn climatic conditions, are often closely related to the posture distribution position and the number of pigs. In previous studies, to determine the recumbency postures of pigs, various classifiers (e.g., linear discriminant analysis (LDA), artificial neural networks (ANNs), and random forest (RF)), were widely applied. For example, Nasirahmadi et al. constructed a linear support vector machine (SVM) model to score different lying postures to explore the relationships between ambient temperatures and pig postures in commercial farming conditions. An accurate classifier effect is more dependent on artificial features and requires high levels of professional expertise for workers. The image processing methods represented by deep learning methods are multi-level feature (shallow and deep features) learning methods composed of simple but nonlinear modules. Each module transforms the representation of a layer into a higher and more abstract representation. Compared with artificial features, the advantages= of deep learning methods is that the model automatically extracts features. For instance, Zheng introduced a faster region-based convolutional network (faster R-CNN) to identify five postures (standing, sitting, sternal recumbency, ventral recumbency, and lateral recumbency), in which the method first extracts the candidate frame of the pig’s posture from the image, and produces a multidimensional feature map using a convolutional neural network (CNN), then performs a secondary correction based on the candidate area to obtain the pig’s position and posture category. In addition, considerable research efforts have been devoted toward constructing different feature extraction and feature fusion models. For instance, the Zeiler and Fergus Net (ZFNet) network was employed by Zheng to identify different sow postures in faster R-CNN [16]. Riekert presented a fused framework that combines a faster R-CNN object detection pipeline and a neural architecture search network for feature extraction [17]. Zhu introduced a refined two-stream algorithm based on a feature-level fusion strategy to automatically recognize pig posture. The model extracts complementarity and "difference" information from RGB and depth images and is more accurate in gesture recognition in lactating sows [18]. Hence, posture detection of individual pigs, on the basis of a deep-learning-based model, has important research potential.

Deep convolutional neural network models and the improved versions are dominant in various fields of computer vision [19]. This is attributed to the development of computer calculation ability, which supports the operation of deep networks. A deeper backbone network enhances the ability to automatically extract features and improves model accuracy. However, detection speed can be slow due to the large amount of calculation required by deeper networks. Moreover, in practice, the demand is urgent for lightweight models to be applied in embedded devices, smartphones, and other low-powered hardware. Given the trade-off between the accuracy and speed of the model, a lightweight and low-latency model, named the MobileNetV1 model, was developed based on a deep separable convolution, and the size of the model was further controlled using two hyperparameters [20]. The MobileNetV2 model is a lightweight network constructed based on inverted residuals with linear bottlenecks [21]. The MobileNetV3 model combines autoML technology and manual fine-tuning for the creation of a lightweight network [22]. Additionally, ShuffleNetV1 applies a channel shuffle operation, allowing the network to use packet convolution as much as possible to improve its calculation speed [23]. ShuffleNetV2 improves upon the design of ShuffleNetV1, and a channel split operation was proposed based on the actual perspective of accelerating the network while reusing features [24]. Lightweight models can be deployed on resource-constrained equipment to automatically detect the posture of individual pigs and analyze pig behavior in a cost-effective manner [25,26].

In this paper, a novel architecture is proposed that combines the efficient channel-wise attention mechanism and lightweight neural networks, which match the resource restrictions on detecting individual pigs with the ultimate goal of 24 h monitoring in an intensive pig farm. The contributions of the proposed method can be summarized as follows:This paper proposes an intensive commercial weaned piglet all-weather posture data set, including multiple pigs, different pig colors, different piglet ages, and complex lighting environments, to benchmark the performance of state-of-the-art models.An optimized shuffle block is proposed based on model structure and parameter statistics. To further improve the accuracy of pig posture detection, we combine the block with the efficient channel attention module. The overall increase in the size of the model is negligible. The results of comparative experiments with state-of-the-art models show that the detection effects of the designed model are more accurate than the corresponding structure. The results show that this method provides higher detection accuracy and reduces the amount of model calculation. Finally, our model deployment plan meets the computer configuration requirements of intensive pig farm production (CPU = i7 and RAM = 8 G).

The rest of this paper is structured as follows. Section 2 presents the experimental environment, data set collection, processing, and detailed methodology adopted in this study. Section 3 shows the results and analysis of the comparison of other models and provides detailed examples. Section 4 summarizes the experimental results and discusses future research direction.

## 2. Materials and Methods

### 2.1. Animals, Housing and Management

All data sets were collected from a pig weaning farm of commercial hybrids located in Yunnan Province, Southwestern China, as shown in Figure 1b,e. The barn had 14 pens (2.2 × 3.5 m) arranged in a row along an outside wall on either side of a central alleyway (28 pens total), with a total of 2404 weaned piglets (large white × long white × Duroc; initial weight = 6.2 ± 0.1 kg). Each pen was equipped with a feeder (4 feeding spaces/feeder) and 2 nipple drinkers on full plastic floors. Floor space allowance was 0.35 m${}^{2}$/pig throughout the study period, excluding the space occupied by the feeder. All pigs had free access to water and adequate feed. A computerized controller and a ventilation system maintained air quality and room temperatures as close to the recommended targets as possible in the barn, as shown in Figure 1f–h. Room temperature at the floor level was recorded in 5 min intervals by a temperature and relative humidity data logger (Prsens, PR-3003-WS-X USB, Tem/RH, China). During the study period, the average room temperature was 26.6 ± 0.5 ${}^{{}^{\circ}}$C (range = 26.6 to 28.5 ${}^{{}^{\circ}}$C) and the average room relative humidity was 67.7% ± 1% (range = 47.8% to 84.7%). Lights in the barn were on for 8 h daily starting from 8:00 GMT. Sufficient feed was automatically placed into the troughs of each pen in the feed line system at 7:00 GMT and 15:00 GMT every day for the pigs to eat freely. The pigs were inspected at least 3 times a day to ensure that pigs were eating and drinking normally, and pigs with diarrhea, fever, or damaged tails were quickly treated.

### 2.2. Data Set

#### 2.2.1. Definition of Pig Postures

Pig postures refer to the association of the location, direction, and crucial parts of the pig’s body. The five classes of pig postures considered in this study are described in Table 1, and the corresponding category cases are shown in Figure 2.

#### 2.2.2. Data Acquisition and Preprocessing

The data sets used in the study were recorded from an infrared camera (BA, BA12–H), as shown in Figure 1a. The bottom of the camera was fixed on the ceiling, the camera lens was tilted down 10${}^{{}^{\circ}}$ counterclockwise, and the angle of the camera was set to capture the posture of all pigs, which needed to be varied to capture the top-view. The cameras were connected via a wireless network video recorder (WNVR) to servers and the images were recorded simultaneously and stored on hard disks. The recorded video with a frame rate of 15 fps was obtained and split into images, and the size of the picture was 2304 in width and 1296 in height. A total of 22,509 images (20,035 for training and validation, and 2474 for testing) were obtained for this data set, which incorporates different piglet color, and illumination conditions, and different intensity levels were selected to provide the training, then the images labeled using the interactive labeling tool LabelMe [27], with a group of three people marking each frame of the posture categories. If one person was not sure about the result of the image marking, the image was deleted from the data set; otherwise, it was added to the data set and saved as JSON files in MS COCO format.

### 2.3. Light-SPD-YOLO Network Architecture

Inspired by the aforementioned classic model, we constructed the SPD-YOLO network model. As shown in Figure 3a, the framework uses the You Only Look Once (YOLO) architecture. The model uses an objection detector to extract feature information from the input image, which directly obtains the pig’s location, posture category, and the corresponding score [28]. The pig posture detector consists of three parts: (1) backbone network, (2) feature fusion, and (3) prediction head.

For the backbone network (BM), a novel symmetric structure is proposed in this paper on the basis of ShuffleNet [24]. The BM structure is shown in Figure 3b: a standard 3 × 3 convolution and a max pool layer are used as the first layer of the backbone network to achieve the downsampling of the spatial dimensions of the input image and to increase the number of channels. Subsequently, continuous stacking of light shuffle blocks improves the ability to extract the posture features of the pig; specifically, the number of repetitions of each branch in the three stages is three, seven, and three, separately. Additionally, we added an ECA module [29] after each stage to improve the learning ability of the network, as shown in Figure 3c.

For feature fusion (FF), the low-level feature information is critical in posture location, but the detailed information is lost due to a long path from low-level to the topmost features maps. Therefore, a path aggregation network [30] was used to retain shallow features in the FF part. In Figure 3d, we demonstrate a continuously performed upsampling operation on the feature map of the last layer and their combination with the feature map of the pyramid layer to obtain new feature maps.

For the prediction head, different scales of feature maps are obtained by the operation of three 1 × 1 convolutions, as shown in Figure 3d. Moreover, a redundant frame is screened to complete the detection of the posture category and the position of the pig based on the NMS operation.

#### 2.3.1. YOLO Principle

In the YOLO algorithm [28], the object detection task is considered a regression problem. The model simultaneously and directly predicts the bounding box and category probability from a given image. The primary principle is shown in Figure 4. Initially, an image is divided into S × S grids, and K bounding boxes are generated for each grid-based anchor box, and C represents the posture category of the pig; each bounding box contains six predicted values: x, y, w, h, confidence, and category probability (Pr). Among them, x, y, w, and h are used to describe the position of bounding boxes, where confidence represents whether the predicted box contains the object and the accuracy of the prediction box, and the pig posture category probability map is simultaneously generated. As stated in Formula (1), confidence is the product of Pr and IOU. If the ground truth falls into this grid cell, Pr (object) takes a value of 1; otherwise, it is 0. IOU is the intersection ratio between the bounding box and the actual ground truth, and the model employs non-maximum suppression to search for inhibiting redundant boxes.
(1)$$\begin{array}{c} Pr\left(Class_{i}|Objection_{i}\right)\times Pr\left(Object\right)\times IOU_{pred}^{truth}\;=\;Pr\left(Class_{i}\right)\times IOU_{pred}^{truth} \end{array}$$
where Pr(Object) is ∈{0,1}

#### 2.3.2. Compression Block

ShuffleNetV2 [23,24] is a computationally efficient and lightweight CNN model. It summarizes the four rules of network design based on the performance of the actual scene, which are as follows: (1) use the same number of input and output channels in the convolutional layer; (2) reduce group convolution operations; (3) model speed is related to model branching; and (4) reduce element-wise operations.

In this paper, the new structure is introduced in the compression block where a 5 × 5 convolution kernel instead of a 3 × 3 convolution kernel in depth-wise, and a 1 × 1 convolution operation is used before and after the depth-wise separable convolution operation have two effects in practice. Either can fuse channel information to compensate for the lack of information due to depth-wise convolution or dimension processing in the model, such as the inverted residual block in MobileNetV2. Therefore, in this paper, the novel symmetric structure is introduced in spatial downsampling operation, including a 5 × 5 depth-wise separable convolution and a 1 × 1 convolution kernel, as shown in Figure 5a,b. Furthermore, the compression block employs a 5 × 5 convolution kernel instead of a 7 × 7 in depth-wise convolution, improving the size and accuracy of the model.

#### 2.3.3. ECA-Net

Lightweight models reduce the number of model parameters through compression modules or model pruning, resulting in feature losses. Previously, researchers have used high-performance attention modules to adaptively adjust the weights between feature channels, such as squeeze-and-excitation networks (SE-Nets) [31], concurrent spatial and channel squeeze-and-excitation in fully convolutional networks (scSE-Nets) [32], and convolutional block attention modules (CA-Nets) [33].

The above model can significantly reduce feature loss, but the dimensionality reduction operation inhibits feature learning. To solve this problem, ECA-Net was applied after the compression block in this study. ECA-Net adopts a local cross-channel interaction strategy without dimensionality reduction and a method of adaptively selecting the size of the one-dimensional convolution kernel *k* to avoid dimensionality reduction operations and effectively adjust the channel feature weight distribution. The insensitive features of the pig postures were assigned lower weights, and the abstract deep features were assigned higher weights, enhancing the network’s representation ability. The architecture of the ECA model is shown in Figure 6.

### 2.4. Evaluation Index

To evaluate the effect of the pig posture detection model, in this study, we adopted the average precision (AP), mean average precision (mAP), and giga floating-point operations per second (GFLOPs), receiver operating characteristics(ROC)curve [34,35]. The ROC curve is created by the true positive rate and the false positive. The formulas for AP and mAP are as follows:(2)$$\begin{array}{cc} Precision & =\frac{TP}{TP+FP} \end{array}$$(3)$$\begin{array}{cc} Recall=TPR & =\frac{TP}{TP+FN} \end{array}$$(4)$$\begin{array}{cc} AP & =\int_{0}^{1}Precision\times Recalldr \end{array}$$(5)$$\begin{array}{cc} mAP & =\frac{AP}{n} \end{array}$$
where *n* represents the pig posture classes. The model parameters and computation complexity were calculated based on the library function in the PyTorch framework.

### 2.5. Network Training

In this study, all models were developed on the PyTorch (https://pytorch.org/, accessed on 6 July 2020) framework. The hardware environment of the study is described in Table 2. The initialization of network weights is critical to the stability of network training; a genetic algorithm (GA) hyperparameter optimization method was used in YOLOV5, named hyperparameter evolution, which is a method to get the optimal parameters by evaluating the cost of fitness [36,37]. The steps of hyperparameter optimization are as follows: firstly, we used a pre-trained COCO hyperparameters list in the initialized hyperparameters. Second, a fitness function was used as a weighted combination of metrics. Among them, mAP@0.5 contributed 10% of the weight and mAP@0.5:0.95 contributed the remaining 90%, with precision and recall absent. The hyperparameter list was obtained after 45,000 iterations, and the final hyperparameter settings are shown in Figure 7. Specifically, in the subgraph, the X- and Y-axis coordinates represent the hyperparameter and the fitness value, respectively; yellow represents the highest concentration, and purple represents the lowest concentration. The final hyperparameter values are shown in the subgraph in Figure 7, including learning rate, SGD momentum, optimizer weight, warm-up epochs, etc.

## 3. Results and Discussions

### 3.1. Performance Comparison of Different Models

Video analysis is one of the main challenges faced by modern pig farms. With 5G communication technology, image processing capabilities and automated detection of pig behavior have also improved. In this paper, an efficient deep learning method was introduced to detect the proportion of the postures of pigs and to automatically monitor changes in pig activity to identify abnormal pig behaviors.

We evaluated the performance of five state-of-the-art networks on the pig posture data set, including faster R-CNN ResNet50, YOLOV3 DarkNet53, YOLOV5 CSPDarknet53, YOLOV5 MobileNetV3-Large, and YOLOV5 MobileNetV3-Small. As detailed in Table 3, faster R-CNN ResNet50 processed the largest number of GFLOPs in all models, and the mean average accuracy rate was 68.6%; YOLOV3 DarkNet53 processed the largest number of model parameters in all models; the mean average accuracy was 81.66%. Compared with the two-stage model, the speed and accuracy of the one-stage model were much better. Additionally, the model parameters and the computation complexity were significantly reduced in the lightweight model. YOLOV5 MobileNetV3-Larger performed the best in terms of the average accuracy of the above model. Compared with YOLOV5 CSPDarknet53, the number of model parameters and the number of calculations were reduced by 26.2% and 37.8%, respectively. YOLOV5 MobileNet-Small is a simplified version based on YOLOV5 MobileNetV3-Large, which further reduces the model parameters. The proposed model possessed the fewest parameters and had the approximately lowest computation complexity among all the models considered. The average precision in detecting STD, LOB LOS, SIT, and MOT was 97.7%, 95.2%, 95.7%, 87.5%, and 84.1%, respectively.

Figure 8 shows examples of pig posture detection using different models in different illumination conditions. Occlusion, illumination, and similarity of posture pose huge challenges for pig posture recognition in commercial fattening houses. Observing the posture detection results in the upper left corner of Figure 8a, note that when part of the body of the pig was missing or blocked by the obstacle, the prediction of the pig’s posture was prone to being missed. Error detection occurred in sitting postures and lying on the belly postures because of the similarity in posture (the posture marked by the red box in Figure 8a is sitting, and the yellow box is lying on the belly). The head of the pig lies closer to the floor, and the front legs are basically hidden in front of the chest. Previous studies showed that adding digital markers on the backs of eight piglets enhanced the detection ability of pig position. The accuracy, sensitivity, and specificity of the model can reach 98.4%, 98.3%, and 95.9%, respectively. In another study, the letters A, B, C, and D were marked on the backs of fattening pigs, and they were monitored by detecting the letters. Different from the experimental scenario—the range of the number of objects in this study was about 9–50 per pen. The marking work is a time-consuming and resource-intensive challenge for the breeder, and the "similarity" of posture cannot be distinguished by external marks.

Under the condition of strong light, as can be seen from Figure 8b, most models missed posture detection (marked in the red frame in Figure 8b). In the study of the posture of a sow, a depth camera was used to solve the influence of light, but the depth camera based on the structured light principle was not selected by pig farmers due to low-resolution reasons. In highly restricted commercial houses, pig farmers are more inclined to use high-resolution cameras to detect as many pigs as possible.

Therefore, on the device, an infrared camera was adopted to mitigate the effects of light. In the model, attention mechanism and feature fusion methods were used to enhance the model’s feature extraction ability: ECA-Net and PANet. Furthermore, the inference time of the aforementioned models in this study was calculated. The result shows the speed of inference is around 528 ms per image for faster R-CNN ResNet50, 321 ms for YOLOV3 DarkNet53, 111 ms for YOLOV5 CSPDarknet53, 127 ms for YOLOV5 MobileNetV3-Large, and 82 ms for MobileNet-Small. In terms of performance, the proposed model is superior to the other models, with 0.358 M model parameters, a MAP of 92.04%, and a speed of 63 ms.

### 3.2. Statistics of Pigs Posture for 24 h

Video data from 24 consecutive hours of a selected pen, including the activities of pigs of different colors and different lighting states, were continuously saved, containing the percentage of the postures, in a .txt file.

The pigs rested during the growth process for 70–80% of the day, with a higher proportion of pigs lying down from 23:00–6:00 GMT, 12:00–1:00 GMT. A few pigs (n = 1–15) ate at various intervals. Another interesting phenomenon occurred during the feeder supplementary period (9:00–10:00 GMT): the pigs were attracted by the friction sound of the feed and the trough structure. Some pigs were awakened and started to eat, which promoted the increase in pig movement. The percentage of the pigs not lying (Figure 9a brown curve) rapidly increase.

From the statistics of the standing posture (Figure 9b), the peak of activity appeared during the feeding period and activity time (15:00–18:00 GMT in the afternoon). When the breeder checks the activity of the pigs, according to the lying down and standing of the pigs, lameness and tail biting of pigs can be detected in advance using the proposed method. In a previous study on pig behavior, an optical flow method was used to detect the changing trend in a pig’s activity amount to intervene in an outbreak of tail biting in pigs. However, the optical flow method requires a large number of calculations and is affected by the light situation. The trend in the posture number changes reflects the changes in the amount of activity in the pigs, which has considerable potential in predicting pig tail biting.

### 3.3. Application in Another Data Set

To further verify the detection ability of the proposed model in this study, the data set of the gestation house of the sow farm was used. Figure 10 shows examples of pig posture detection using different models in different detection situations. The figure shows that the pig’s body was obscured by the railings or whole postures were not visible, which poses challenges for the task of pig posture recognition. Occlusion is the reason for the missed detection shown in Figure 10a, noting that when part of the body of the pig was missing or blocked by the railing, the prediction of the pig’s sitting posture and lying on the belly posture was prone to errors because of the similarity in these positions (Figure 10b). The average precision in the proposed model for detecting postures is 94.8%.

### 3.4. Limitation of Proposed Model

Our research has several limitations. First, in this research, we focused on assessing behavior by detecting the positions and static postures of multiple pigs, but the performance degradation of high-level behavior (e.g., mounting, explore) detection was consistent with previous work, using deep learning to detect pig postures (SEO), and the latter pays more attention to the classification of posture. It is worth noting that, in their studies, a single complete pig posture was segmented from the whole picture, which helps balance the number of categories in the data set. One possible reason for the performance degradation of high-level behavior is the temporal characteristics and spatial characteristics of continuous postures is not introduced.

Previous studies showed the conversion method of single sow posture [38]; in their work, the timestamp was in the posture detection task, which helped break down the action clips in the video stream of interest. Another study used a refined two-stream algorithm based on feature-level fusion; the method extracted the complementarity and difference information of RGB and depth images.It is worth noting that the number of pigs in our work caused a dilemma pertaining to the implementation of the above algorithm. The task of continuously tracking multiple pigs is a huge challenge.

Second, although the mAP of the proposed model is satisfactory, the APs of the sitting or mounting categories are lower than expected. One explanation is that the distribution of the data set is unbalanced (Figure 11) Therefore, in future work, data enhancement is needed by segmenting a single pig pose [39].

Third, our application of posture detection was only used for weaned piglets. Pig pose detection in other periods is challenging due to complexity of posture. Therefore, it is necessary to collect data sets from birth to slaughter.

## 4. Conclusions

Automatic detection of individual pigs is one of the main challenges faced by intensive pig farms. Our work focuses on detecting pig poses and positions on resource-constrained platforms, automatically monitoring pig activity changes to predict abnormal behavior. It can detect the following postures of pigs: standing, sitting, lying on the side, lying on the stomach and riding. The average accuracy rate is 92%, the model parameter is 0.358 M, and the GFLOPS is 1.2. Compared with the most advanced lightweight models, the performance of this model is better. In terms of improving accuracy, a moderate increase in the size of the depth convolution kernel can improve the performance of the model with a small number of parameters. Secondly, the ECA-Net attention network is used to improve the expression of feature channels. The model performs well in monitoring the location of pigs and has high accuracy.

In the pig industry, changes in posture reflect the activity and resting behavior of pigs. Continuous monitoring of the posture behaviors of pigs could help reflect abnormal behaviors of pigs and promptly encourage managers to intervene. For example, there may be problems in the sitting pig postures (i.e., lameness). The reduction in the proportion of stance in a pig herd can indicate the occurrence of infectious diseases in the pig herd. In addition, our experimental results show that images of different scenes are necessary for model training, to improve the generalization ability of the detection model. Therefore, it is necessary to further obtain data sets of different pig houses and different camera angles to ensure the performance of the new pen layout. One advantage of deep learning is that by labeling new data sets and fine-tuning the model to these training data, the applicability of the new pen layout can be improved iteratively.

## Figures and Tables

**Figure 1 sensors-21-08369-f001:**
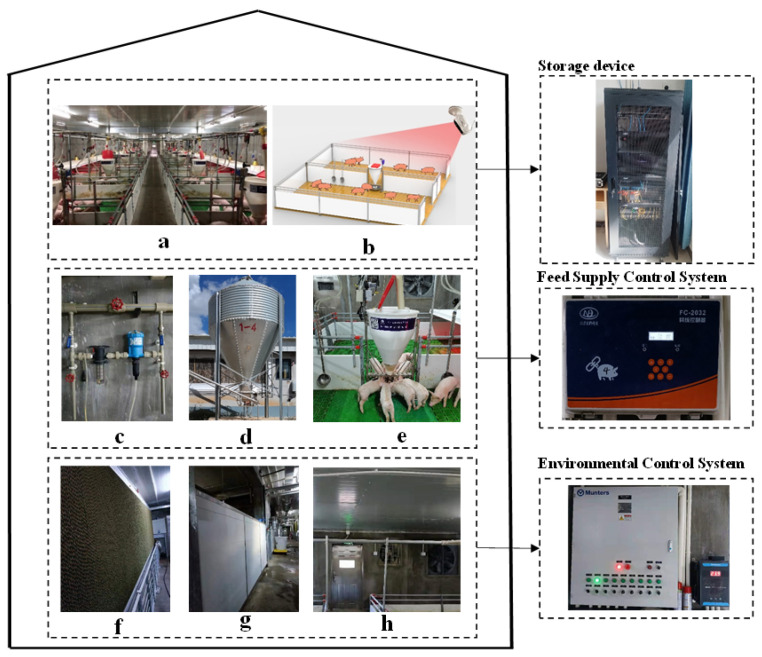
Overall dimensions of the pig housing. (**a**) Inside the pig house. (**b**) Camera shooting position. (**c**) Water supply pipeline. (**d**) Feed tower. (**e**) Faucet and feeding equipment. (**f**) Wet curtain. (**g**) Baffle. (**h**) Fan.

**Figure 2 sensors-21-08369-f002:**
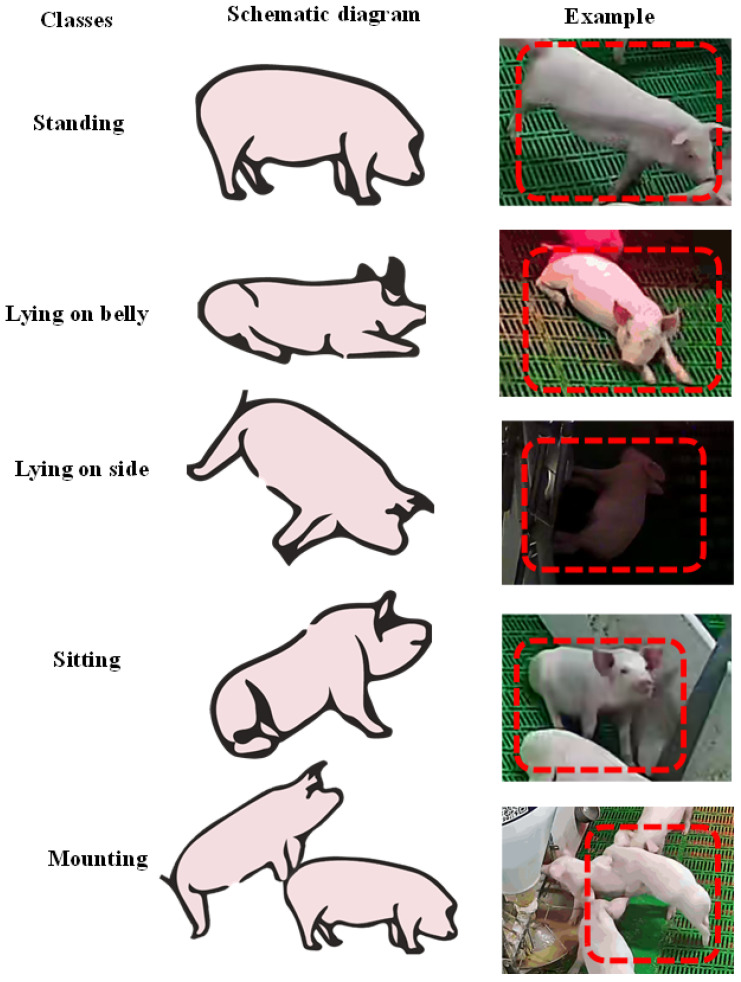
Definition of pig postures.

**Figure 3 sensors-21-08369-f003:**
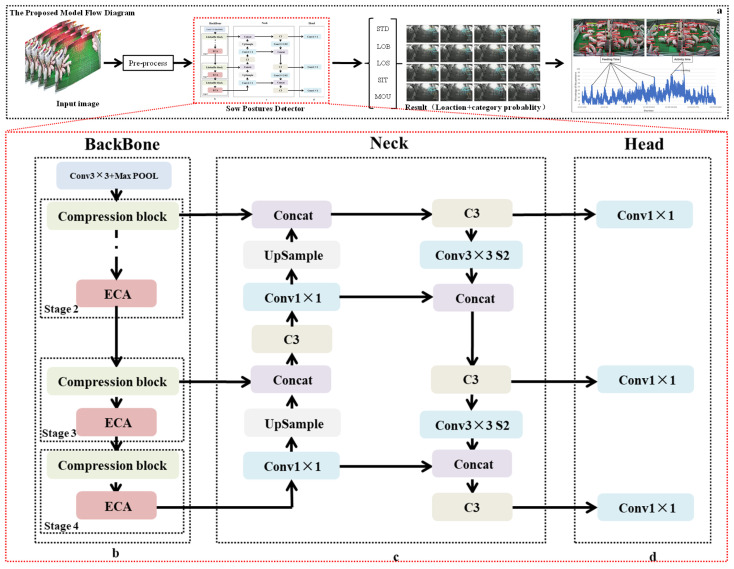
Pig posture detection flowchart. (**a**) Specific process of the model; (**b**) Backbone of the model; (**c**) Neck of the model; (**d**) Head of the model.

**Figure 4 sensors-21-08369-f004:**
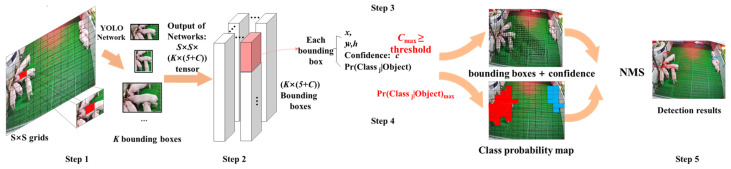
The primary principle of YOLO.

**Figure 5 sensors-21-08369-f005:**
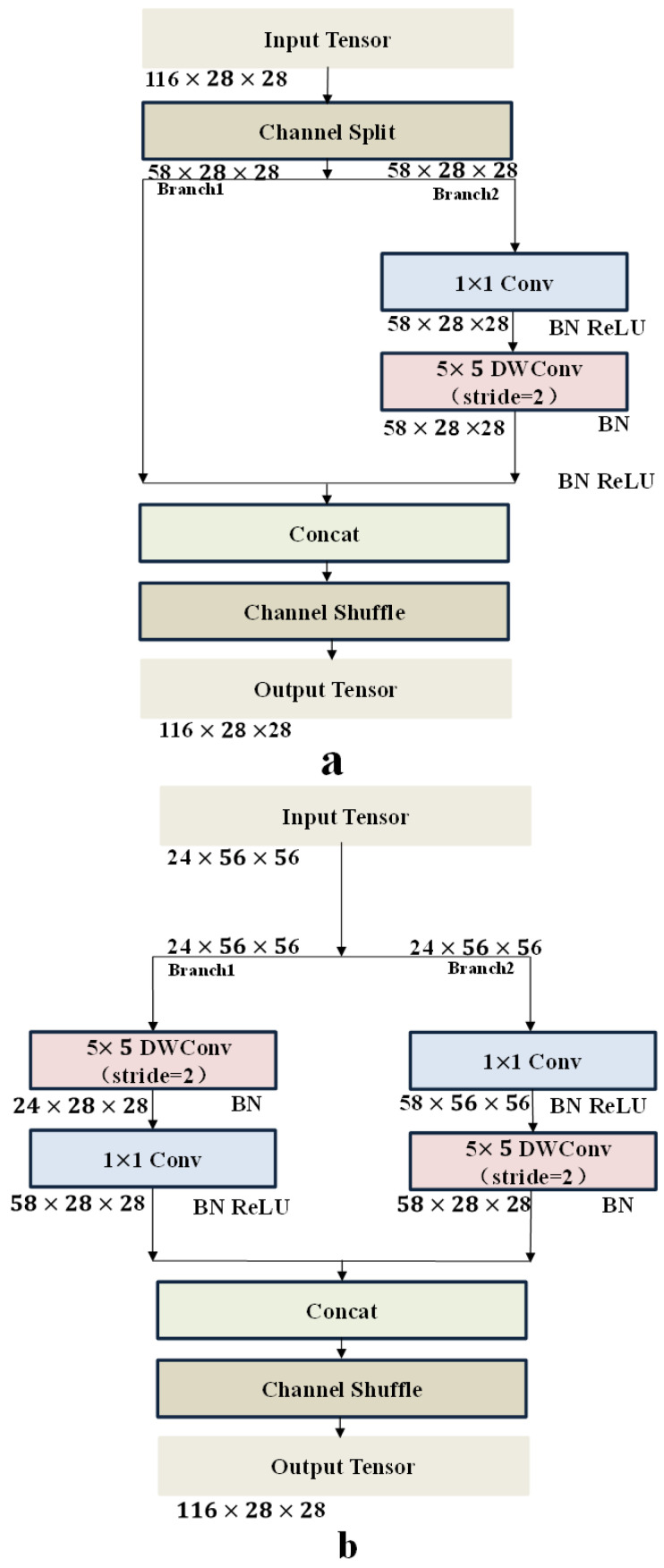
Structure of the compression block. (**a**) basic compression block unit and (**b**) basic compression block unit for spatial downsampling.

**Figure 6 sensors-21-08369-f006:**
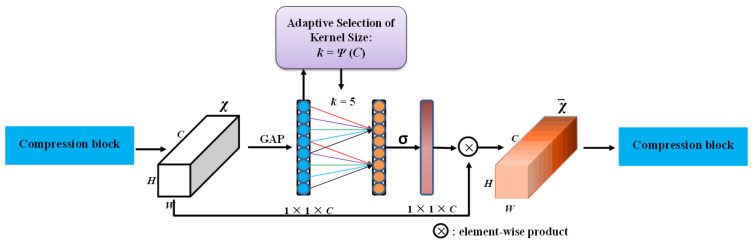
ECA model architecture. GAP means global average pooling, k is adaptively determined via a mapping of channel dimension C [33].

**Figure 7 sensors-21-08369-f007:**
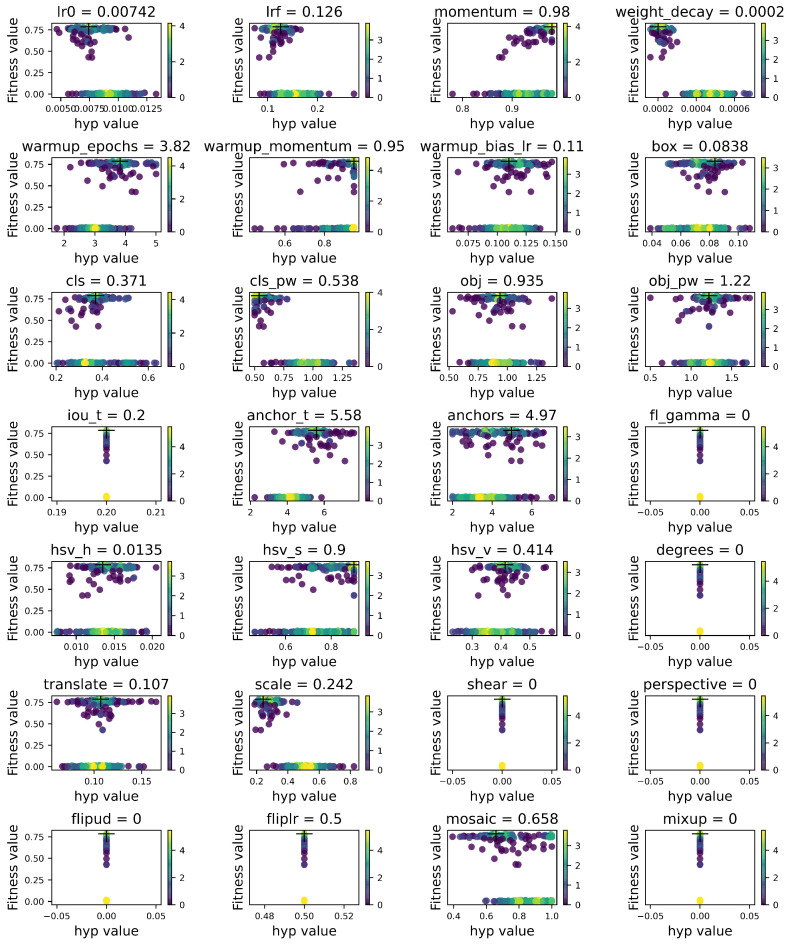
Hyperparameters obtained using a GA.

**Figure 8 sensors-21-08369-f008:**
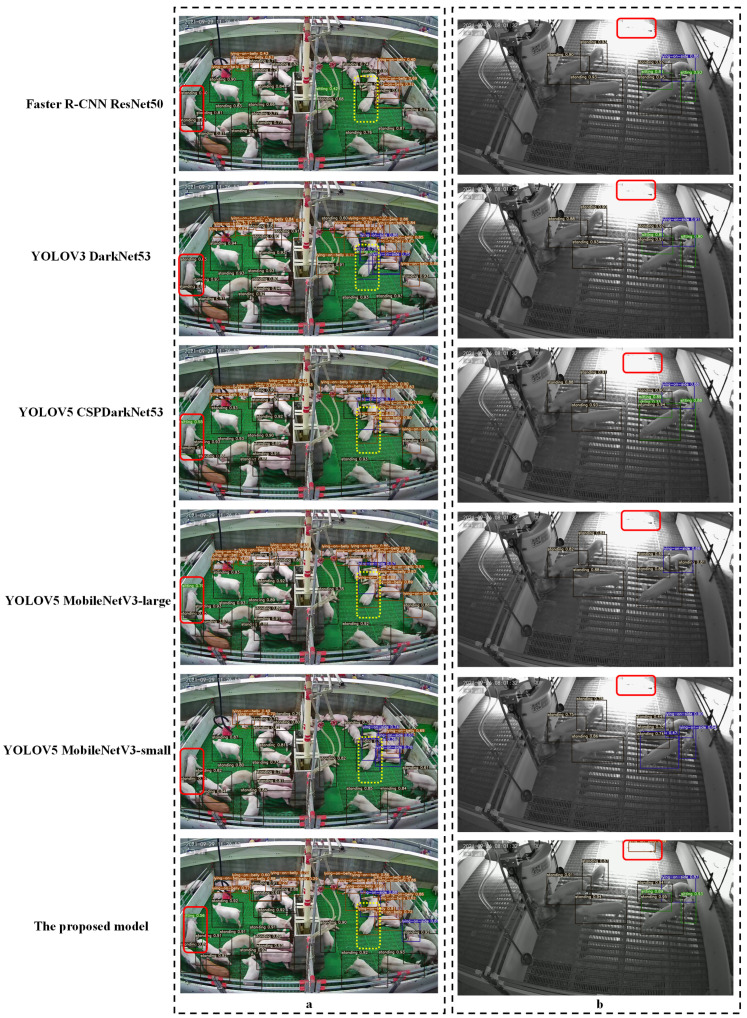
Examples of pig postures detection based on seven models. (**a**) example with lighting; (**b**) example without lighting.

**Figure 9 sensors-21-08369-f009:**
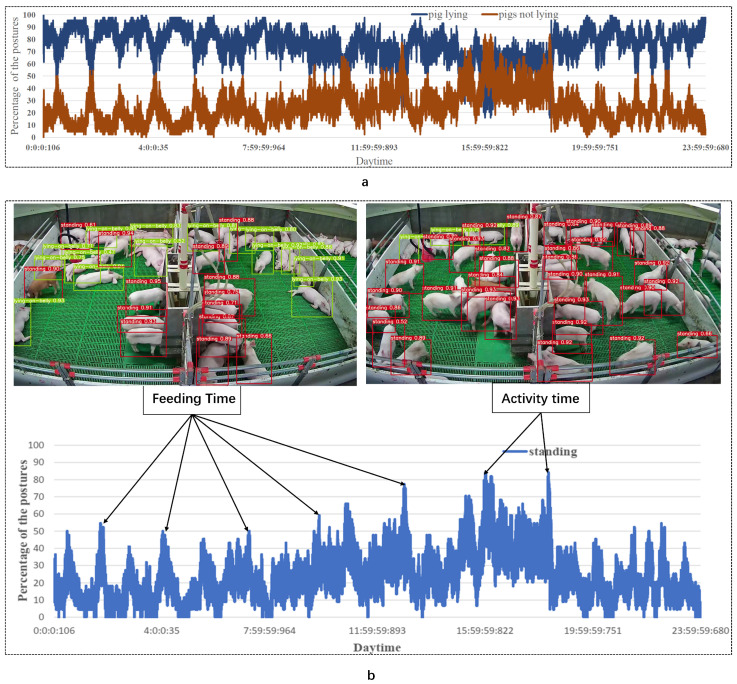
Analysis of pig postures. (**a**) Statistics of lying behavior of pigs in the nursery house for 24 h; the blue curve represents pig lying down, and the brown curve represents the pig not lying down; (**b**) Standing statistics of pigs in nursery house for 24 h.

**Figure 10 sensors-21-08369-f010:**
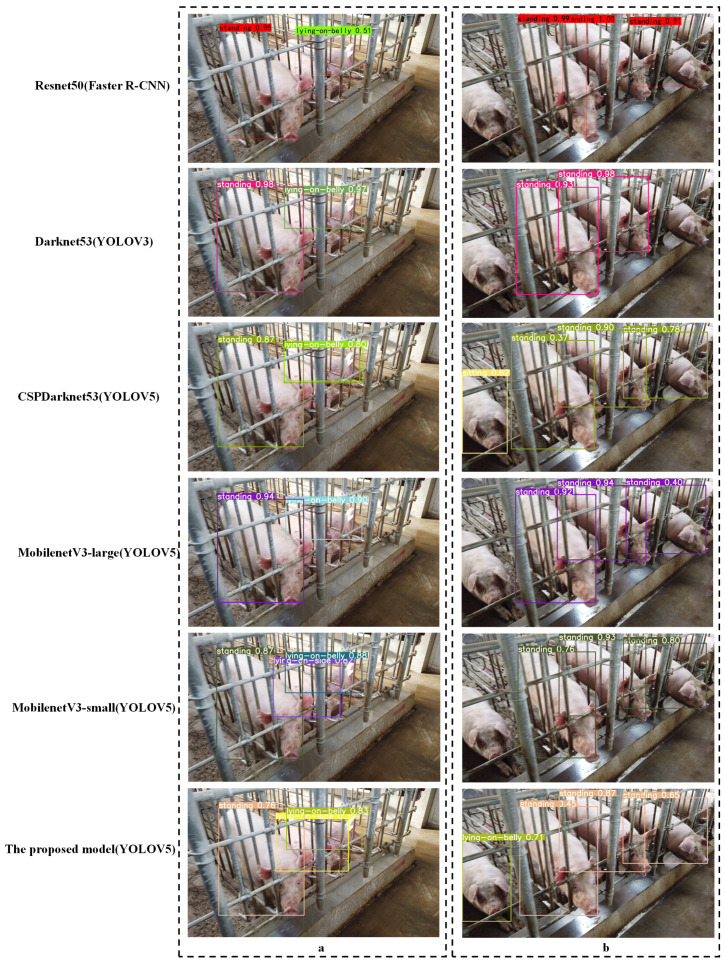
Examples of sow postures data set for a restricted pen. (**a**) example of missed inspections; (**b**) of error detection.

**Figure 11 sensors-21-08369-f011:**
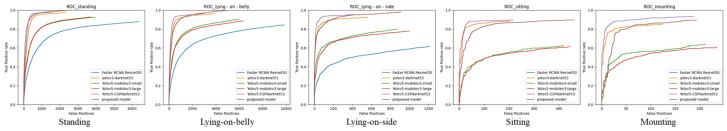
ROC curve for each category (standing, lying-on-belly, lying-on-side, sitting, mounting).

**Table 1 sensors-21-08369-t001:** Definition of pig postures.

Pig Posture Category	Abbreviation	Posture Description
Standing	STD	Upright body position on extended legs with hooves only in contact with the floor [16].
Lying on belly	LOB	Lying on abdomen/sternum with front and hind legs folded under the body; udder is obscured [16].
Lying on side	LOS	Lying on either side with all four legs visible (right side, left side); udder is visible [16].
Sitting	SIT	Partly erected on stretched front legs with caudal end of the body contacting the floor [16].
Mounting	MOT	Stretch the hind legs and standing on the floor with the front legs in contact with the body of another pig.

**Table 2 sensors-21-08369-t002:** Experimental hardware environment.

Configuration	Parameter
CPU	Intel(R) Core(TM)4210
GPU	NVIDIA GeForce RTX 2080Ti
Operating system	Ubuntu 18.04 system
Accelerated environment	CUDA11.2 CUDNN 7.6.5
Development environment	Vscode

**Table 3 sensors-21-08369-t003:** Performance comparison of different models.

Detector	Backbone	mAP	Parameters	GFLOPS	fps(CPU = i7)	AP				
						**Standing**	**Lying-on-Belly**	**Lying-on-Side**	**Sitting**	**Mounting**
Faster R-CNN	ResNet50	68.6%	28.306 M	909.623	528 ms	83%	72%	69%	60 %	59%
YOLOv3	Darknet53	81.66%	61.539 M	154.9	321 ms	90.7%	90.1	85.9%	77.3%	64.3%
YOLOv5	CSPDarknet53	81.96%	0.706 M	16.4	111 ms	90.4%	90.4%	84.8%	75.2%	69%
YOLOv5	MobileNetV3-large	78.48%	0.521 M	10.2	127 ms	90.7%	90.1%	83.8%	63.3%	64.5%
YOLOv5	MobileNetV3-small	76.76%	0.357 M	6.3	82 ms	91.2%	90%	85.2%	63.1%	54.3%
YoloV5	Light-SPD-YOLO (proposed model)	92.04%	0.358 M	1.2	63 ms	97.7%	95.2%	95.7%	87.5%	84.1%

## Data Availability

Not applicable.

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
