# Peer review of "Posture Detection of Individual Pigs Based on Lightweight Convolution Neural Networks and Efficient Channel-Wise Attention"

_sensors, 2021, doi:10.3390/s21248369_

Round 1
Reviewer 1 Report
In the paper, a lightweight pig posture detection algorithm is proposed for the real-time detection of five representative pigs postures: standing, lying on the belly, lying on the side, sitting, and mounting. The results show that the algorithm’s average precision in detecting standing, lying on the belly, lying on the side, sitting, and mounting is 97.7%, 95.2%, 95.7%, 87.5%, and 84.1%, respectively, and the speed of inference is around 63 ms per postures image.
The topic is very interesting. The manuscript structure is well. The manuscript has some of its technical merits. It should fall into Journal Sensors. However, several questions have to pay attention to:
- In the abstract, the authors introduced many backgrounds about their issue. It can be moved to the introduction section. The section can be improved.
- In section 1, the authors mentioned, “At present, lightweight networks have achieved……”. “However, those have rarely been used for sow posture recognition tasks.” It may not be a good reason. The authors should highlight why to study your issue and give the benefit from the study.
- In the last paragraph, section 1, the authors add a whole structure description of the paper. It will be better.
- In section 2.4, the authors provided evaluation indexes, such as ap, map, gflops. However, those indexes can’t reflect the dataset unbalance. Recommend the authors should provide a ROC curve for each category.
- In section 2.5, the author introduced hyperparameter evolution. The tune hyperparameter is a critical step in network training. The authors need to provide how to obtain the hyperparameter in detail.
- In section 3.1, the authors mentioned “accuracy, sensitivity, and specificity.” How to obtain it? What about other comparing networks? Recommend the authors provide one table about the results.
- The discussion section can be improved. The author has not pointed out the disadvantage of the proposed algorithm or a comparison of enough previous literature. Recommend the author should extend your discussion.
- The author might enrich your conclusions. It might also suggest future research.
Hopefully, this will help in the revision of the manuscript.
Author Response
Dear reviewers:
Thank you for your comments concerning our manuscript. Those comments are all valuable and very for revising and improving our paper, as well as the important guiding significance to our researches.
Please see the attachment.

Reviewer 2 Report
Posture Detection of Individual Pigs is the focus of the research topic. This topic is interesting because automatic recognition of individual pig’s postures based on vision imaging systems is becoming more and more common on commercial pig farms.
In this paper, a lightweight pig posture detection algorithm is proposed for the real-time detection of five representative pigs postures: standing, lying on the belly, lying on the side, sitting, and mounting.
The proposed model combines a lightweight block and efficient channel-wise attention models to enhance performance while reducing model parameters. Compared with the state-of-the-art lightweight models, this proposed model performs better in terms of average precision, fewer model parameters and lower computation complexity.
Some typos are addressed in the following:
1. In page 10: In the 3rd paragraph, Error detection occurred in sitting postures and 260 lying-on-belly postures because of the similarity in posture (the posture marked by the 261 red box in Figure 8a is sitting, and the green box is lying on belly).--> There is no green box?
2. In page 14: Support vector machine: SVM --> Not RF
Author Response

(The authors gave the same response as above.)

Round 2
Reviewer 1 Report
The authors have improved their manuscript according to most of the points.